# Representations of Hulk Hogan in the 1980s: Christianity, Masculinity, Xenophobia

Conor Heffernan 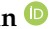

School of Sport, Ulster University, Belfast BT1 6DN, UK; c.heffernan@ulster.ac.uk

**Abstract:** In 1984, professional wrestler Hulk Hogan defeated the Iron Sheik to win the WWF Heavyweight Title. Thus marked the birth of 'Hulkamania', a near-decade-long period when Hulk Hogan (real name Terry Bollea) crossed over into American popular culture. In the following years, Hogan battled a series of proxies for America's enemies, from the Soviet rival in Nikolai Volkoff, Iranian sympathizer Sergeant Slaughter, and the Japanese sumo wrestler Yokozuna, among other opponents. More importantly, Hogan appeared on American talk shows, the front of magazines, had his own children's cartoons, and marketed workout devices, toys, food, and a host of other ephemera. Existing in a liminal space between sport and entertainment, professional wrestling allows athletes/performers far more opportunities to cultivate messages and meanings through their bodies. Using film, wrestling magazines, and wrestling broadcasts, this article argues that Hogan's body and his use of his body were paramount to his success. More than that, the use of his body embodied ideals about American masculinity. It embodied all-American strength, an ability to succeed no matter the odds, and a fierce Christian patriotism. Hogan was one of the biggest stars of the 1980s, inside and outside of sport. His body and its representation are thus worthy of study.

**Keywords:** professional wrestling; Hulk Hogan; American History; masculinity; popular culture

## 1. Introduction

A camera cuts to 'Mean' Gene Okerland, the well-known interviewer and announcer for the World Wrestling Federation, visiting Hulk Hogan's 'private gym' where Hogan was preparing his body for an upcoming match against King Kong Body. As Hogan runs through a set of stiff leg deadlifts, replete with athletic tape surrounding his ribs, the audience is reminded of a 'brutal' attack Bundy and his cadre of villainous friends inflicted on a defenseless Hogan on the WWF's *Saturday Night's Main Event* show on 1 March 1986. Hogan's training partners, a physician, and a fellow wrestler monitor Hogan's health before Hogan tells Okerland that 'for five and a half weeks I've been with this thing [his back pain]'. Disclosing the pain he is in, Hogan claims that he cannot let his fans down and has to train through the pain. The segment continues with Hulk Hogan strapping a hundred-pound dumbbell around his neck and doing chins ups. Gene Okerland is incredulous at Hogan's strength and determination, imploring Hogan not to hurt himself. Hogan continues to tell Okerland and the audience about what he is going to do to Bundy once they face off in a cage match at the upcoming Wrestlemania event (WWF 1986b). For audience members, the segment was presented in Homeric terms, the ailing hero against the villainous monster, willing to risk all to fight for good. For scholars, the segment is notable for the primacy placed on Hogan's body. It is simultaneously presented as injured yet strong. It was a site of heroism but also of upstanding American manhood (Smith 2014). The purpose of this article is not to analyze a throwaway segment on *Tuesday Night Titans*, a wrestling show from the 1980s. Rather, the purpose of this article is to explore the role of the muscular, white body in the success of 'Hulkamania'. Titled after wrestler Hulk Hogan (real name Terry Bollea), 'Hulkamania' was the name given to the popularity and frenzy which surrounded Hogan in professional wrestling and American popular culture (Chard and Litherland 2019, p. 29).

As Mondak (1989) has cited, wrestling in twentieth-century America underwent three periods of intense popularity prior to the famed 'Attitude Era' of the mid to late 1990s. They were the 1930s, the 1950s, and the 1980s. Hogan, in particular, was critical in the sport's popularity during the 1980s. Hogan appeared on MTV, the American sitcom A-Team, cereal commercials, and a host of other sites of consumption. Later works on Dwayne 'The Rock' Johnson have noted his intertextual significance—spanning across film, television, social media, etc. Hogan experienced a similar ubiquity in the 1980s (Ward 2019). He was loved, and at times reviled, within the industry for this dominance as the WWF's main wrestler. In 1985, for example, Hogan was voted the Best Babyface (the name given to the heroic figure in wrestling) and the Most Overrated Wrestler by the *Wrestling Observer*, then a fringe but nevertheless influential magazine for diehard wrestling fans (Wrestling Observer Yearbook 1985, p. 17). In both instances, his profile within the sport was cited as often as his physique. Studying Hogan's popularity and influence, I argue that Hogan's manipulation of his physical body clearly underscored ideals of American masculinity that were themselves grounded in discourses of popular Christianity and sport established in the nineteenth century. This is executed by first examining the role of the body in professional wrestling, with reference to Pierre Bourdieu's theory of habitus, and subsequent Bourdieusian work on physical capital. While Bourdieu's work has previously been applied to professional wrestling (Chard and Litherland 2019), the theory of habitus has largely been neglected. Following this, the article discusses the rise of 'hard bodies' in American popular culture during the 1980s before examining Hogan's ability to capitalize upon his own muscular frame in mock battles against America's rivals.

## 2. Studying Professional Wrestling and Physical Capital

Professional wrestling is, by its nature, a body-focused endeavor. Operating on the nexus between traditional sport and pantomime theatre, wrestling in the United States was presented, officially, as a legitimate sport until the end of the 1980s when chairman of the World Wrestling Federation, Vince McMahon, conceded that it was pre-planned entertainment (McAuliff 2012). Part of the sport's body-centric ethos can be traced to the late nineteenth century when many wrestlers were also budding physical culturists (a term given to prototype fitness celebrities and coaches). While many of these wrestlers were legitimate athletes, competing against one another, pre-planned matches began to emerge. Indeed, one of the most high-profile wrestling bouts of the pre-World War 1 period between George Hackenscmhidt and Frank Gotch lasted several hours and included biting, eye gouging, and a host of other nefarious tactics (Lindaman 2000). Such brutality explains, in part, why wrestlers began to pre-script their matches. Equally important, as Stadel (2013) has argued, was the sport's crossover with entertainment at an early stage. Nevertheless, McMahon's admission in the 1980s that wrestling was pre-planned still sent shockwaves through the industry. At that time, McMahon was the most authoritative voice in wrestling due to his near monopolization of the industry. McMahon's concession, based on his desire to avoid paying the athletic commission fees required of legitimate sports, was not, of course, the first time wrestling had been 'exposed' as fake in the twentieth century, but it was the most definitive example. Despite this, McMahon and his wrestlers have continually stressed that while the results may be pre-determined, the athleticism and danger found in wrestling are all too real. Put another way, how the body moves, presents itself, and hurts itself defies the fakery otherwise found in wrestling. Arguably, an overcited piece of wrestling scholarship is French philosopher Roland Barthes' article on the subject for *Mythologies* (published originally in 1957, although written first in 1954). Covering a regional French wrestling bout, Barthes noted the excess embedded in professional wrestling. More importantly, Barthes discussed the connection between the wrestler's body and the characters they played within the ring

> The body of the wrestler is seed of his character, and, like the Commedia dell' Arte, his appearance is a perfect predictor of his actions . . . (Barthes 1957)

This should come as no surprise. While some high-minded individuals may trace professional wrestling to Greco-Roman times, professional wrestling of the kind experienced by Barthes had its origins in eighteenth- and nineteenth-century fairs, Vaudeville stages, and music hall shows. In the United States, where Hogan made his fame, professional wrestling's primary origins, as detailed by John W. Campbell (1996), were in the carnival scene. As Campbell made clear, there came a point that the carnival wrestlers, competing against each other every evening, realized that cooperating with one another, rather than physically fighting each other, would protect their bodies and their bank balances in equal measure. This is not to say that wrestling from the late nineteenth century was entirely pre-determined. Indeed, well into the 1920s, legitimate forms of wrestling (i.e., those whose outcomes were not scripted) proved to be incredibly popular. In many instances, athletes were divided into 'shooters' (those who could and would legitimately wrestler) and those who were entertaining characters that lacked the same physical skills. Where at one point wrestling cards held a combination of real and scripted, by the 1930s, wrestling was predominantly pre-scripted as a form of entertainment. By that point, various companies in different parts of the United States offered wrestling on a weekly basis. The wrestling landscape of the United States from the 1930s until the 1980s was a series of established 'territorial' companies operating in different regions, loosely aligned under the banner of the National Wrestling Alliance. This structure was eventually dismantled by Vince McMahon's WWF as McMahon began buying out these companies to establish his company as the main wrestling outlet. Hulk Hogan proved to be a boon in gaining popular acceptance for this strategy (Laine 2019). It is at this point worth commenting that the fakery and deception embedded within the sport has undergone significant scholarly analysis. Within wrestling itself, the term 'kayfabe' is used to denote the ways in which the performers and managers create illusions and/or deceive the fanbase. Within the industry, kayfabe often implies a one-way relationship whereby the fans (often referred to derogatively as 'marks') are easily duped (Moon 2022). In fact, researchers such as Reinhard (2019) in this area have consistently stressed the symbiotic relationship between performer and fan when it comes to the creation and maintenance of kayfabe. As Hill's (2015) work on fans and anti-fans in wrestling has shown, many fans actively 'buy in' to kayfabe and help to maintain the illusion of the performance.

Regardless of whether wrestling was predetermined or not, the physiques of the wrestlers themselves have long been 'an object of fascination and desire.' Wrestling's upsurge in popularity at the turn of the twentieth century coincided with the rise of the physical culture movement. Defined by Michael Anton Budd (1997, pp. x–xii) as a 'late nineteenth and early twentieth-century concern with the ideological and commercial cultivation of the body,' physical culture has been credited by many as a precursor to modern bodybuilding and health cultures. Importantly, for the purposes of this article, many physical culturists also engaged in wrestling, including George Hackenschmidt. As Broderick Chow's (2015) work on Hackenschmidt made clear, Hackenschmidt's relationship with his body was critical in both his wrestling and physical culture careers. Even individuals such as Eugen Sandow, famous for his role in popularizing bodybuilding, began his career as a wrestler. This was also true for Bernarr Macfadden, one of America's first fitness entrepreneurs. The connection between physical culture/body cultures and wrestling that continued throughout the twentieth century and into the present one is that hard, muscular, bodies have tended to be privileged over others.

It is at this point that a discussion of *habitus* and physical capital must be made. Coined, and in the case of physical capital, inspired by French sociologist Pierre Bourdieu, these concepts provide a means of unpacking the importance of the flesh in professional wrestling. Noble and Watkins (2003, p. 522) previously understood *habitus* as 'the dispositions that internalize our social location and which orient our actions'. Simplifying this, *habitus* can be defined as the manner in which individuals negotiate their lives through learned practices and behaviors, some of which are embodied. There is no one unifying *habitus* for individuals and, depending on their social world, individuals may display different *habitus* depending

on their situation or 'field' (in school, in the gym, at work, etc.) (Bourdieu 1977). Previous wrestling scholarship has already successfully utilized the concept of Bourdieusian fields to discuss the success and difficulties that wrestlers have entering into other realms of popular culture. Some, such as Dwanye 'The Rock' Johnson, have seamlessly built profiles outside of wrestling in film and television. Others such as Hulk Hogan, the focus of this article, have struggled. While some of the skills and capital Hogan acquired in wrestling were advantageous for other fields, Chard and Litherland (2019) stress that the rules and tensions of one field are never the same in another. Surprisingly, despite some excellent works on the performance of the body in wrestling, *habitus* has yet to be fully utilized within the context of professional wrestling outside of some cursory, but nonetheless excellent, works.

Due to its focus on embodied practice, sociologists have used this concept to describe the unique bodily movements and skills needed in sports and pursuits such as boxing, ballet, tennis, and a range of other activities. In professional wrestling, and this point was discussed by Broderick Chow (2015), countless repetition of movements is used to ingrain in athletes the ability to execute moves and mannerisms as if they are perfectly natural. Loic Wacquant's (2004) work on boxing showcased the body's ability to learn a boxing *habitus* and, in time, be able to respond spontaneously to attacks with pre-learned reactions. The professional wrestler's skillset and in-ring mannerisms can be viewed as another form of *habitus*, and indeed, although the sport is pre-scripted, many of the moves seen in the wrestling ring are decided in the moment between the two wrestlers, and based on the crowd's reactions. In the case of Hulk Hogan, his wrestling style revolved around a core set of seemingly natural movements and mannerisms exhibited on a regular basis (a leg drop, shaking of the fingers, 'hulking up'—soon to be discussed). Importantly, these movements are 'full body' in that it was not just the movement of Hogan's muscles and limbs that enticed audiences, but also his ability to emote pain, shock, anger, etc. Speaking on his own chosen finishing move, late 1990s and early 2000s wrestler Mick 'Mankind/Cactus Jack/Dude Love' Foley revealed that he specifically chose a move in which the television cameras could see both his and his opponent's facial expressions (Holder 2022). Whereas some wrestlers decide to meticulously plan every move prior to a match, as Hogan's contemporary Randall Poffo ('Macho Man') was famed for doing, others preferred to 'call it in the ring' based on their interactions with their opponent and, more importantly, the crowd's reaction (Zarka 2022). In this way, a wrestler's *habitus* does not exist outside of themselves but rather in a multi-agent dynamic which extends to their opponent, the crowd, television cameras, and, potentially, referees.

Examining the various ways in which individuals achieve and maintain success, Bourdieu discussed the various forms of capital available. These included economic capital (stocks, bonds, property, etc.) but also more ephemeral forms such as cultural capital (accrued by individuals who possess a societally desirable trait), social capital (tied to one's economic class), and symbolic capital (tied to traits such as honor, prestige, or recognition). Returning to Chard and Litherland (2019), they noted that alongside the cultural capital (not to mention economic capital) that Hogan built during his wrestling career, he built significant symbolic capital as an individual capable of miraculously recovering from injury and exhaustion in the ring to defeat his opponents. Initially, Bourdieu viewed physical capital (the value attached to one's body) as a subfield of cultural capital. This has been disputed by subsequent scholars in the 1990s and 2000s who have strongly argued for the need to view physical capital as worthy of its own classification (Shilling 2003). Work by fair and on ballet dancers has made clear the ability individuals have to convert their physical capital into social or economic capital. Put another way, it is possible to use one's body to achieve financial and social success precisely because one can mold their bodies to fit desirable societal ideals. The wrestler George Hackenschmidt, studied by Chow (2015), is one such example. Operating as a weightlifter, wrestler, and physical culturist, Hackenschmidt was able to advance his social station to previously unheralded levels.

When Terry Bollea re-entered the WWF in the early 1980s as Hulk Hogan, both the wrestling scene and the broader American context were becoming fascinated with large, muscular bodies. In the WWF, Vince McMahon's ideal body type for his athletes was noticeably strong and muscular. While wrestling had previously operated on the basis that variety was best in terms of wrestlers, McMahon was a self-confessed 'body guy' (Heffernan and Warden 2022). Thus, tall and large wrestlers (be they muscular or rotund) began to take precedence in his company. To advance, one needed to be muscular. This contrasted with previous decades in which obviously athletic, but not necessarily muscular, wrestlers were given the top billing. Hogan was not the first wrestler to display an over-the-top muscularity (that accolade likely goes to Billy Graham—a wrestler who inspired Hogan's own gimmick), but Hogan's was the first to enjoy a truly crossover appeal with this kind of physique.

This drive for muscularity later hurt the WWF when a federal trial into steroid use by his athletes in 1994 made clear the pressure many felt to achieve his ideal. The trial, which attempted to prosecute McMahon for distributing anabolic steroids to his wrestlers (Heffernan and Warden 2022), hurt the WWF and Vince's short run bodybuilding federation (The World Bodybuilding Federation) as it forced McMahon to institute strict drug testing in shows. Without large muscles, it was difficult to advance to the company's upper echelons, a point noted by certain wrestlers in the trial itself (Kaelberer 2003). By the time of Hogan's fame, the WWF was America's largest wrestling company which also influenced smaller wrestling organizations (Kaelberer 2003). As discussed in the next section, American society was likewise becoming fascinated with large and muscular men.

## 3. Reagan Hard Bodies and Hogan

In 1977, *Pumping Iron* became a smash hit at American box offices. Focused in part on a youthful Arnold Schwarzenegger, the documentary was the first time that bodybuilding and bodybuilding cultures had truly broken into the American mainstream. What was previously viewed suspiciously, as a deviant if not homosexual endeavor, was repurposed by Schwarzenegger and others as heterosexual and mainstream. The success of *Pumping Iron* played a significant role in normalizing the muscular male physique outside of sporting endeavors (Roach 2008). While athletes' bodies had long been the subject of media attention in the United States, the bodybuilder's body was markedly different. The bodybuilder wore their muscular frame like a uniform throughout the day. The body was inescapable and highlighted their pursuit of muscle, often divorced from a sporting context. As Alan Klein's (1993) critical work from the early 1990s made clear, this new era in bodybuilding was deeply concerned with the pursuit of muscle for muscle's sake, in the belief that the hard outer body of the bodybuilder reflected messages and values about their masculinity, work ethic, and ability to withstand pain. Importantly, and as Klein reflected, the muscular body became seen as the ideal physique for American men, especially white American men determined to showcase their masculinity. Building this body necessitated countless hours in a gymnasium, a demand many believed highlighted broader masculine traits of self-discipline and work ethic. Returning to Bourdieuisan approaches, the muscular and large physique came with an immense prestige. Physical capital mattered. When it came to promoting *Pumping Iron*, director George Butler held an event at the Whitney Museum of American Art wherein the bodybuilders posed on plinths akin to tableau vivants as art critics and journalists wandered the room (Lowry 1976). What began as a publicity stunt was, in fact, a critical moment in highlighting the new social value attached to muscularity and bodily control.

It was not just male bodies either. In 1980, the International Bodybuilding Federation, the premier bodybuilding federation in the United States, hosted the Ms. Olympia competition, a women's bodybuilding show. While women had engaged in bodybuilding practices prior to this point, and at times had been celebrated for it, the 1980s was a period when muscular women were also enjoying mainstream attention. One obvious example of this is American photographer Robert Mapplethorpe's photographs of Lisa Lyons, the first winner

of the Ms. Olympia competition. Whereas Lyon's photos were initially a cult phenomenon, Hollywood too took notice. Perhaps the most famous examples of this was found in James Cameron's 1991 box office success *Terminator 2: Judgement Day*, which include a scene of female protagonist Linda Hamilton doing pull-ups and displaying the muscularity of her biceps (Richardson 2016).

Throughout American life in the 1980s and early 1990s, be it bodybuilding or jogging, the fit physique came to be celebrated as an expression of one's self-determination, work ethic, and social value (Latham 2015). This ethos of individual struggle and hard work fit neatly into an American zeitgeist infected with Ronald Reagan's own version of rugged individualism. As Reagan's biographer Robert Dallek (1999, p. 4) later wrote, Reagan was imbued from an early age with the idea that 'energy and hard work' were all that were needed to succeed. This drove Reagan but also became part of his political and economic strategy as president of the United States. Elected in 1981, Reagan's tenure in the United States has previously been associated with the demonization of those seeking welfare and the celebration of individual successes (Kohler-Hausmann 2015). Physical activity was part of this process, a point Benjamin Rader (1991) previously noted. Under Reagan, physical activity became an acceptable form of leisure, as it cultivated these traits. The interplay between Reagan's administration and the new American zeal for muscle building came to a head in 1983 when *Parade* magazine detailed the serving president's workout routine. Predictably, the interview was a call for individual self-action and discipline from quotes criticizing those who 'have a problem sticking to their exercise routine' to those noting Reagan's self-discipline: 'I pass up the pancakes and sausage in favor of cereal and fruit, skim milk and decaffeinated coffee' (Reagan 1983). While it may seem trite to discuss the president's eating and training habits, Jeffords made clear that Reagan was a master of self-construction. He tapped into what Rader (1991) referred to as America's desire for self-sufficiency and strenuosity during this period. Trickling down from the president himself was the zeal for physical activity and the linking of American masculinity (more accurately white American masculinity) to one's body. In Hollywood, a new genre of action films emerged centered on the hard, white, muscular bodies of American men fighting for their countries, families, or their beliefs (Jeffords 1994). This body was combined with a dogmatic, while simultaneously xenophobic, belief in American values and self-sufficiency. Sport and media were perfect conduits for this message (Kammen 1993). Oftentimes, the two were conflated. Take, for example, the 1980 American ice hockey victory over the Soviets, the 'miracle on ice' in which the Americans came from behind to win the match. The victory struck such a chord in the American zeitgeist that it was turned into a movie the very next year. As Chad Seifred's work on melodrama found, the victory in the film was presented in part as due to the innate American attitude to 'never say die' and maintain self-belief in the face of much stronger opponents (McDonald 2012). One could argue that the *habitus* comeback sequence of Hulk Hogan embodied a much broader American fascination with melodrama and redemption. After all, American pop culture could point to *Miracle on Ice*, the *Rocky* franchise, *Hoosiers*, *The Karate Kid*, and a host of underdog stories (Vandello et al. 2016).

Turning towards Hulk Hogan, the WWF astutely tapped into this linking of American ideals and American muscle in two ways: first in pre-recorded training videos and promos; second with Hogan's in-ring posing and theatrics. The previously mentioned video segment of Hogan recuperating in the gym prior to a fight was not the only time the WWF relied on footage of Hogan training in the gym to promote an upcoming match and, in doing so, to reiterate Hogan's masculine courage. At times fans were even given a point of comparison between the 'regular' man and Hogan. In the 1984, Hogan tag teamed with 'Mean' Gene Okerland against the 'villainous' pairing of George Steele and Mr. Fuji. As part of the build up to the match, the WWF showed vignettes of Hogan critiquing Gene's lazy ways. At 5 a.m., Hogan meets Gene at his home where he is shocked to find him smoking and about to indulge in pancakes. Next, the duo head to a running track where Hogan spouts inspirational lines (including 'eye of the Hulkster') as Gene

wheezes behind him. Later vignettes show the two in a gym as Gene struggles to match Hogan's strength and power (WWF 1986a). The skits, done for comedic effect, nevertheless showcased Hogan's distinction from the 'typical', non-fighting American and his daily commitment to a lifestyle of strength in much the same way that *Pumping Iron* did for Arnold Schwarzenegger (Lowry 1976). While these skits were primarily for wrestling fans, later videos had a much broader audience. The sketches inspirationally end Hogan telling Gene that 'they said we couldn't do it but with all of St. Paul ... behind us, we're going to do it' (WWF 1986a).

In 1985, Vince McMahon and the WWF launched a new pay-per-view event, *Wrestlemania*, which has gone on to be the sport's most iconic yearly event. At the time, the WWF had begun to break through to the mainstream through a partnership with MTV and an influx of celebrity appearances and endorsements, including that of Cyndi Lauper (Pro Wrestling Illustrated, Anon 1986). One of the showpiece spectacles of *Wrestlemania* was a tag team match between Hulk Hogan and Laurence Tureaud (better known as the A-Team's Mr. T) and two WWF wrestlers Paul Orndorff and Roddy Piper. The rivalry itself had been several months in the making, with crossovers between Mr. T and the WWF and Hulk Hogan and the A-Team (where Hulk Hogan appeared in a single episode). In the buildup to the match, the WWF filmed several vignettes of Hogan and Mr. T training in weightlifting and boxing gyms, as well as in public. Still available on the *WWE Network*, the clips celebrate Hogan and Mr. T's self-sacrifice and hard work. Viewers at home were shown Hogan and Mr. T strenuously exercising in public gymnasiums, rallying each other to lift heavier weights, in front of an adoring public (WWF All Star Wrestling 1986).

Next, the video switches to Hogan and Mr. T training in a boxing gym together, a scene made stranger by the fact that unlike Mr. T, Hogan had no discernible boxing experience whatsoever. Nevertheless, it reinforced the muscular and strong physiques of both men. As discussed, American society put great emphasis on these sorts of bodies as shorthand for determinism and manliness. The two thus exhibited their physical capital for all to see. Finally, and in homage to *Rocky III* (debuted in 1982), the two are shown running down the street with members of the public struggling to keep up (WWF All Star Wrestling 1986). Given that Mr. T was Rocky's rival in the film, it is impossible not to view the montage as an attempt to gain some form of crossover appeal (Chard and Litherland 2019). It also, unintentionally, highlighted the pervasiveness of the 'hard body' ideal across multiple platforms.

Footage of the two men training was interspersed with videos of 'Mean' Gene Okerland interviewing Hogan and Mr. T. Glistening with sweat, they preached about their hard work, incomparable strength, and determination to defeat their rivals. Their physical appearance, musculature, and marshalling of their own bodies was presented as the only way to take a heroic stance against the evil Orndorff and Piper (WWF All Star Wrestling 1986). Much in the same way that Rocky Balboa overcame his foes through sacrifice and training montages, so too did Hogan and Mr. T. This was a reoccurring theme in Hogan's career, regardless of whether he was shown training alone, or with his fellow 'heroes'. The WWF repeatedly relied on footage of Hogan in the gym, be it with Mean Gene, Hillbilly Jim, or a host of other wrestlers. If Hogan and Mr. T's montages focused on Hogan's self-discipline and his literal embodiment of heroic ideals, later vignettes introduced another component, his steadfast Christian beliefs.

Soon after his rise to fame in the WWF, discussed in the next section, Hogan introduced fans to his 'demandments', which he promoted as the cornerstone of any American man's behaviors. They were to 'train, say your prayers and eat your vitamins' (Mazer 1998, p. 47). A later fourth 'demandment' was added—believing in yourself. Speaking primarily to a younger audience (Hogan often divided the older wrestling community's opinion), the overtones with Christian commandments are obvious. Further hammering home his steadfast Christianity was the crucifix that Hogan wore around his neck. The crucifix even became part of one of Hogan's biggest storylines in the WWF: his feud with André René Roussimoff or André the Giant.

As one of the largest men to step inside a wrestling ring, André had been a showpiece attraction in the United States for two decades, but near the end of the 1980s, his mobility and health had seriously waned. This was due to his untreated acromegaly. In a now famous interview segment, hosted by Roddy Piper, André confronted his former friend Hogan and challenged him for Hogan's World Heavyweight Championship. As tensions heat up, André grabs Hogan and rips his vest off completely. In doing so, André also tore Hogan's crucifix from his neck, which left a small but noticeable cut on Hogan's chest. Hogan capitalized on this moment by later claiming that the contract for their eventual fight was not 'signed in ink . . . but signed in blood', referring to his crucifix (WWF Prime Time Wrestling 1987). Although unintentional, the image of Hogan's blood draped across his chest is a further illustration of the way in which his body added to his persona. In other areas of wrestling, performers intentionally cut themselves ('blading') to safely but dramatically add blood to a particular match (Hadley 2016). Done effectively, blading can add to the drama of a match and heighten concerns around the punishment suffered by a performer. Although Hogan's cut was minor, it added to the betrayal of his former friend André.

Hogan and André's feud ended when Hogan—in a moment the WWF disingenuously claimed was the first time anyone had ever body slammed André—defeated André at *Wrestlemania*. While that match is iconic and, as some have argued, helped propel the WWF's popularity even further, it is the aftermath that is of interest here. In 1988, the WWF *Saturday Night Main Event* (February 5) featured a two-minute training montage of Hulk Hogan in the gym. Run to promote Hogan's later match with André that evening, the montage shows Hogan running through his routine of squats, chin ups, bench presses, and bicep curls. Dressed in all black, albeit in a vest, a noticeable inclusion around Hogan's neck is a gold crucifix like the one André previously ripped from his neck. At one point, the camera zooms into Hogan's upper body to focus on the crucifix while he grunts his way through a set of bicep curls. The messaging is simple, and blunt. Hogan is once more training to defeat an unconquerable foe. His body, fueled by his adherence to his own 'demandments', represents his seriousness.

Here one sees the juxtaposition between Hogan's muscles and a distinctly American form of 'muscular Christianity'. Originally a nineteenth-century phenomenon begun in the British public school system, muscular Christianity came to be simplified by the Latin phrase *mens sana in corpore sano*, a 'healthy mind in a healthy body'. The belief that a strong constitution was illustrative of a strong mental and spiritual life came to be a motivating factor in the nineteenth-century sporting movement of Britain and the United States (Putney 2009). The Reagan eras of 1980s were, as Steven Miller made clear in a work on American evangelicalism, a time when the president of the United States appeared to be guided by a strict interpretation of the Christian Bible. Reagan's morals, and more importantly, his work ethic, were depicted as being hugely symbolic and even inspirational for a rejuvenated conservative Christian strain in American society (Roof 2009). The muscular Christian ideal of the nineteenth century—which celebrated a Weberian Protestant work ethic—was reinvented during the 1980s. As Benjamin Rader (1991) argued, the strive for self-sufficiency during this period echoed the underlying ethos of the muscular Christian ideal—that hard work, as found in one's bodily strength and appearance, was tantamount to some form of personal good. Put another way, in the 1980s, the body became a visual signifier of one's social wealth. The fit body acted as a proxy for the religious undertones from previous century—it showcased sacrifice, self-discipline, and devotion. Returning to Bourdieu, one sees the physical capital of Hogan's body being paired with the symbolic capital of his Christian ideals (as encapsulated by his crucifix). During the match that evening, Hogan is defeated by André but only through a convoluted storyline involving a corrupt and imposter referee (who is the twin brother of the originally scheduled referee).

Accepting that montages of Hogan training in the gym, in preparation for an upcoming fight, were well worn tropes, it is also worth noting Hogan's own skill in using his body to

elicit responses from the crowd. On entering the wrestling ring, Hogan typically tore his own shirt off and moved into a bodybuilding 'most muscular' pose wherein the biceps and shoulder muscles are hunched forward and tensed to make the body appear as freakish as possible (Mazer 1998, p. 47). Borrowing from performance studies, the regularity with which Hogan performed this action transmitted, without words, the message that his was an impressive and masculine physique (Chow 2015). The fin de siècle physical culturists studied by Chow (2015) often operated as both performers and living statues, so too did Hogan. Entering the ring, Hogan was adept at using bodybuilding poses to stress his physical capital. He would hit front double bicep poses, most muscular poses, and poses designed to accentuate the size of his back (French 2014). At other points, to amplify the audience's reaction, Hogan would pose and then cup his hand to his ear to goad the audience into cheering even louder for him. Muscles and performance combined to stress Hogan's strict adherence to the masculine ideals so celebrated during the Reagan regime.

While also adept with a microphone, Hogan's skill lay in the use of his muscular body to convey messages. This was seen in Hogan's famed 'hulking up' sequence. During Hogan's matches in the WWF there typically came a point when it appeared that Hogan's opponent was on the verge of victory. As Hogan lay motionless on the ground, hurting under the blows from his opponent, a moment would come where Hogan would rise slowly from the canvas and lift himself up to one knee. As a now shocked opponent redoubled their efforts, striking Hogan again and again, Hogan would wag his finger and slowly rise to his feet. As his opponent grows even more worried, Hogan would throw them into the ropes and greet them with a boot to the face as they ran back towards him. The sequence ended with Hogan bouncing off the ropes and delivering a leg drop across the opponent's neck/upper chest for the victory (Shields 2010, p. 37). This sequence was so common during the 'Hulkamania' era that once Hogan initiated it, the crowd would begin to loudly cheer in anticipation of the eventual victory. In this, Hogan perfectly encapsulated the hard body ideal (Vostel 2022). His 'comeback' *habitus* was in line with *Miracle on Ice*, *Rocky*, *Hoosiers*, and any number of 'great' American victories depicted in popular media. Much in the same way that Wacquant (2004) developed his boxing *habitus* through countless hours spent in the boxing gym, Hogan's 'hulking up' *habitus* became second nature to him and his fans. His ability to move his body in a repetitive, almost robotic fashion signaled a sequence that fans knew from beginning to end. His muscular frame was also to singlehandedly defeat his foes, sustain great punishment in return and still, despite his pain and tiredness, emerge victorious. Returning to the self-sufficiency preached by Reagan and his acolytes, also found within Hollywood films, Hogan embodied these ideas in theatric ways.

## 4. Hogan, Embodiment and Xenophobia

How Hogan built, used, and displayed his body was of utmost relevance in building his popularity as a wrestler and cultural figure. Equally, if not more, important were the opponents that Hogan faced in the WWF. Working in the dying embers of the Cold War, and a time of substantial American intervention in foreign lands, Hogan's opponents could be read as a crib sheet for America's perceived enemies. In much the same way that action movie stars proved their heroics in Hollywood by defeating Soviet counterparts, some of Hogan's biggest feuds came against those who reveled in their hatred for America. The WWF/WWE, as multiple scholars have asserted, has often played on crude and outdated stereotypes as villainous characters ranging from antisemitic tax collectors (Benton 2015) to militant black nationalists (Koh 2022) or over-sexualized 'jezebel' female characters (Corteen 2021). Hogan's early celebrity was in part fueled by his victory over one such character, Hossein Khosrow Ali Vaziri ('The Iron Sheik'), in January 1984. At the time the Sheik was the WWF's champion and top villain following his victory over American wrestler Bob Backland and a series of anti-American promos. Coming to the ring with an Iranian flag, and vocally disrespecting America, the Sheik–Hogan match in 1984 was presented as one of America against her enemies (Rahmani 2007). Following the 1979 Revolution in Iran, and the installation of Ayatollah Ruhollah Khomeini as leader, U.S.–

Iranian relations had been fraught. In the years preceding the Hogan–Sheik match, there had been economic sanctions by the U.S. against Iran, a hostage crisis involving U.S. citizens, and allegations of terrorism against Iran by the Reagan administration. The Sheik appeared to stand defiantly in the United States as a visual representative of Khomeini's regime. In the match, the Iron Sheik locked Hogan into the 'Camel Clutch', a fearsome submission move, which fans had been conditioned to think was inescapable (Rahmani 2007). As the Sheik continued to apply pressure to Hogan, Hogan began to power back up. Although the 'hulking up' montage had not yet been perfected, Hogan slowly rose from the ground and threw the Sheik into a turnbuckle before delivering his own finishing move for the victory. The crowd, as Chard and Litherland (2019, p. 29) later noted, broke into chants of 'USA, USA, USA'. This moment marked the first time Hogan won the WWF heavyweight championship and has been credited by many as the 'birth of Hulkamania,' and was in many ways the template for several of Hogan's later feuds. The veracity of these feuds was built upon a pro-America nationalism that was decidedly xenophobic in its expression. In these matches, Hogan served as a proxy not just for 'good over evil' but, more pointedly, for America over its enemies. That Hogan often came to the ring draped in an American flag, proclaimed that he was fighting for the United States, and professed his love for America was a blunt, but effective, way of aligning Hogan, and his body, to American ideals.

The alignment between Hogan and the American flag is a particularly interesting case study as later feuds, such as that with Josip Hrvoje Peruzović ('Nikolai Volkoff'), reflected American Cold War politics. In 1985, and on WWF's *Saturday Night Main Event*, Hogan and Volkoff had a 'flag match' wherein Hogan wrestled with an American flag in his corner and Volkoff wrestled with a USSR flag in his corner. Managed by long-time villainous character 'Classie' Freddie Blassie, Volkoff's character revolved around being an American-hating Soviet who had teamed with Hogan's other enemy, the Iron Sheik, at several points during this period (WWF Saturday Night's Main Event 1985). Supposedly fighting for their own country's pride, the match encapsulated Hogan's pro-America/xenophobic character. After the match, when Volkoff had been thoroughly defeated by Hogan following his 'hulking up', Hogan turned his attention to the Soviet flag being held aloft by Freddie Blassie. Hogan headbutts the Soviet flag before taking it to a corner of the ring wherein he began to shine his shoes with the flag. He finishes by throwing the flag on the ground and spitting on it, to the approval of the packed stadium. In two senses, Hogan used his body as a pro-America tool (WWF Saturday Night's Main Event 1985). The first was, rather obviously, in defeating the evil Volkoff, after it seemed inevitable that Volkoff would cheat to win. With victory assured, Hogan slowly but purposively begins to desecrate the Soviet flag using his head, feet, and saliva. If blood is used by wrestlers to amplify drama within a feud, saliva is often depicted as the ultimate insult, and indeed, spitting on another wrestler is often the catalyst for intense fighting. Looked back on with modern eyes, the pageantry is kitschy. Viewed another way, the event saw Hogan intentionally use his actions to goad the American audiences to a near fever pitch of anti-Soviet anger.

While Hogan's opponents were not always anti-American villains, it was a reoccurring theme within his 'Hulkamania' tenure. One of the last opponents to fall into this category, Robert Remus ('St. Slaughter'), was likewise defeated by an angry Hogan defending the American flag. Wrestling Hogan at *Wrestlemania VII* in 1991, Slaughter was previously known as an American-loving staff sergeant whose fame spread outside of wrestling when he was co-opted to be part of the G.I. Joe franchise. In late 1990, Slaughter returned to the WWF following a protracted absence. Rather than playing a pro-American hero, he instead proclaimed himself to be a pro-Iraqi sympathizer (Rahmani 2007). Done during the Gulf War, Slaughter was presented as a traitor to the American flag and all that it entailed. While the WWF's popularity was waning during this period—*Wrestlemania's* venue had to be moved due to a lack of ticket sales—the company still attached its hopes on Hogan defeating the anti-American villain (Rinehart 1998). Going into the event, Hogan and the WWF fans were angered by stories that Slaughter had burned an American flag

during an anti-American tirade. Speaking with 'Mean' Gene Okerland, Hogan responded to Slaughter's alleged traitorous act.

> Mean Gene, let me tell you something . . . if Sergeant Slaughter has gone so far to deface Old Glory, to deface the red, white and blue, brother, I don't care if it's legal or not brother . . . just like Saddam Hussein's reign over Kuwait, brother, it's [Slaughter's reign] gonna be *only temporary*. (WWF Royal Rumble 1991)

While the flag 'desecration' was never televised, it did enough to enrage fans. In Hogan's interviews with Gene Okerland, and in subsequent vignettes, the message was always the same: Hogan would defend America's honor by physically beating his enemy. What differentiated Hogan–Slaughter, in part, from the Hogan–Volkoff match previously discussed was the 'crimson mask' (the term used to describe a bloodied face) Hogan wore during the match (Chow and Laine 2014)—in the case of Hogan, his bloodied and tired face, especially when the wrestler appears to be losing the match (Mazer 1998, p. 164). In a typical playbook, Slaughter appeared to be on the verge of winning before the bloodied Hogan 'hulked up' and won the match. Fans were eventually left with the image of the blood-stained Hogan waving an enlarged America flag as the ultimate symbol of patriotism.

Hogan's xenophobia did not just extend to America's Cold War enemies, but also African and Asian nations. This is perhaps best seen in his 1987 feud with James Arthur Harris ('Kamala'). Billed as the Ugandan Giant, Kamala donned a loincloth, wore a necklace of bones around his neck, and wore what was advertised as African war paint on his face (Campbell 1996). Hogan and Kamala feuded at several points during the 1980s, although it would be wrong to suggest that Kamala was an iconic Hogan opponent. Nevertheless, the two did get involved in several well received feuds during the mid-to-late 1980s. It is the buildup to a steel cage match (wherein the ring is enclosed by a steel cage) in 1987 which is of interest in this article. Interviewed prior to the match, Hogan appeared on television wearing his own war paint on his face in mimicry of Kamala. Warning Kamala that the steel cage would help to separate the line between the man and the ultimate beast, Hogan continued to speak about 'natives in the mumbo jumbo pot' before proclaiming that Hogan would prove that he, alone, was the 'ultimate beast' (WWF Live Event 1987, 11 January). Wearing his new war paint, Hogan then mimicked, if not mocked, Kamala by uttering a guttural noise and sniffing the camera. Reinforcing Kamala's racialized gimmick, Hogan went one step further by proclaiming that he, himself, could beat Kamala by embracing a form of beastliness that could surpass Kamala (WWF Live Event 1987, 11 January). That Hogan continued to wear his crucifix during the promo reinforced that despite a temporary move to beastliness, he still held his Christian ideals. The promo was racially tone deaf at best, if not outright offensive, but nevertheless marked a final way in which Hogan's body was used to convey harmful stereotypes about other races or other ethnicities. It could be used simply as a heroic device, as a proxy for American exceptionalism or American racialism. Blood, muscles, and paint were three obvious methods.

## 5. Conclusions

Hogan was, undoubtedly, the face of professional wrestling in the 1980s. He main-evented the most popular shows, appealed to those outside the sport, and had one of the most inventive income streams ever seen by a wrestler. Hogan was presented as the WWF's heroic figure, the man who defended America's honor at every possible turn and someone who could inspire the WWF's audience to be the best version of themselves. His appeal, and indeed his messages, centered on a very clear version of white American masculinity which was muscular, clean living, and brave. Hogan was, in essence, a conduit for the broader American idea of white masculinity which permeated Hollywood film in the 1980s. At the peak of his power, Hogan could command the audience's attention with nothing more than the wag of a finger. His storylines with other wrestlers and his matches served to reiterate Hogan's importance and intrigue for wrestling fans. The WWF used Hogan as a proxy for America itself—an America which refused to back down, which fought foreign aggressors, and an America deeply divided against foreigners and people of color. It was

an unquestioned sense of Americana espoused by mainstream media and even the Reagan administration. Hogan's popularity stemmed from his ability to embody and transmit these messages in a range of settings.

In 2002, Hulk Hogan returned to the WWF (since renamed WWE) after a protracted period with a rival organization, WCW. During his tenure with WCW, Hogan's character evolved from the heroic American man who always triumphed to an egotistical villain who referred to himself as 'Hollywood Hulk Hogan'. Upon returning to the WWF, Hogan reverted to his former type and, for a time, even played a masked character known as Mr. America. While his second run with the WWF did not enjoy the same crossover into popular culture, his brief tenure provided a nostalgic return to the 1980s when over-the-top Americana dominated wrestling. The wrestler's body, as Roland Barthes famously critiqued, is a conduit for theatrical messages about good and evil, right and wrong. Although widely regarded by hardcore wrestling fans as a limited wrestler at best, Hogan's genius and fame came from his ability to use his body in the way Barthes described. Shaking his muscles, wagging his finger, donning face paint, wearing jewelry or bandages, Hogan's physique was a palimpsest for broader American concerns and aspirations surrounding the country's position on the global stage, Christian muscularity, American masculinity, and the ideal body. This explains, in part, why Hogan and the WWF proved so successful in 1980s America. Through words, movements, and images, it was a conduit for a dominant thought in the zeitgeist and, more importantly, broke out from 'mere' wrestling into popular culture. His performances—and they were performances—were a synergy between man, movement, and messaging. Hulkamania was more than just words—it was flesh.

**Funding:** This research received no external funding.

**Data Availability Statement:** No new data was created.

**Conflicts of Interest:** The authors declare no conflict of interest.

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
