# Peer review of "Representations of Hulk Hogan in the 1980s: Christianity, Masculinity, Xenophobia"

_arts, 2022_

Round 1

Reviewer 1 Report

I quite enjoyed this essay. It gives a nice overview of the importance of bodies in professional wrestling, with a nice snapshot of Hulk Hogan’s career and key narratives in Rock N Wrestling America. I think it is always worth restating how central professional wrestling is to global popular cultures, and the ideologies that are contained therein.

The main issue with the essay is that the engagement with existing scholarship on professional wrestling is, at best, quite cursory. There is now quite an extensive 30+ year history of exploring some of the key issues outlined in the essay, particularly relating to the relationship between representations and gender, class, and nationality; the wider history of professional wrestling, particularly in the 1980s and WWF’s expansion; and the “work” of pro wrestler’s, their bodies, and the forms of exploitation that might be contained there.

Pro wrestling of course reproduces certain values and ideologies, but much of the scholarship since the 90s have explored not just the stereotypes, but the role of audiences in co-producing these meanings, sometimes resisting them, the relationship to wider celebrity and commercial culture industries, and the complicating factor that post-ironic kayfabe readings might have in muddying some of the assumptions. Given the essay’s rightful critique of the over-referencing of Barthes, it would be helpful if some of the debates and discussions here were put into dialogue with the wider field.

There are whole sections of history or discussion about the WWF that pass without reference, so as a reader it’s a struggle to get a sense of where those claims are being drawn from: for an industry like pro wrestling, steeped in hearsay and gossip, I think it’s important to reference scholars who have cited and used primary sources. Likewise, given that Chard & Litherland cover some of these issues in their own essay (including some engagement with Bourdieu), I’d possibly expect more than a single citation illustrating the key theoretical or conceptual divergences and departures.

To give you a scope of the field I’ve outlined some texts that haven’t been consulted and which might help develop some of the central points and arguments. These don’t all need to be read and cited, but they will help the essay to develop some of the nuances of the argument as they are presented.

History

Hilton, L. (2007). Ringside: A history of professional wrestling in America.

Litherland, B. (2018). Wrestling in Britain: Sporting entertainments, celebrity and audiences. Routledge.

Stadel, L. (2013). Wrestling and cinema, 1892–1911. Early Popular Visual Culture11(4), 342-364.

Warden, C. (2020). " Queer music-hall sport": all-in wrestling and modernist fakery. Modernism/modernity27(1), 147-164.

Politics, stereotypes and nationalisms

Benton B (2015) Lamination as slamination: Irwin R. Schyster and the construction of antisemitism in professional wrestling. The Journal of Popular Culture 48(2): 399–412.

Koh, W. (2022). A new day for Hulk Hogan: Celebrity selves and racial diversity in contemporary professional wrestling. European Journal of Cultural Studies25(2), 778-796.

Foy, M. (2018). The ballad of the real American: A call for cultural critique of pro-wrestling storylines. The Popular Culture Studies Journal, 173.

Levi, H. (2005). The mask of the luchador: wrestling, politics, and identity in Mexico. In Steel chair to the head (pp. 96-132). Duke University Press.

Mondak, J. J. (1989). The politics of professional wrestling. Journal of Popular Culture, 23(2), 139.

Jenkins, H. (2005). ‘‘Never Trust a Snake’’: WWF Wrestling as Masculine Melodrama. In Steel Chair to the Head (pp. 33-67). Duke University Press.

Bodies, capital and labour

Corteen, K. (2018). In plain sight–examining the harms of professional wrestling as state-corporate crime. Journal of Criminological Research, Policy and Practice4(1), 46-59.

Chow, B., & Laine, E. (2014). Audience affirmation and the labour of professional wrestling. Performance Research19(2), 44-53.

Gardener, B. (2016). Spatial maneuvers: geographies of power and labor practices in professional wrestling’s territorial era. In Critical Geographies of Sport (pp. 223-235). Routledge.

Hill, A. (2015). Spectacle of excess: The passion work of professional wrestlers, fans and anti-fans. European Journal of Cultural Studies18(2), 174-189.

Smith, R. T. (2008). Passion work: The joint production of emotional labor in professional wrestling. Social Psychology Quarterly, 71(2), 157-176.

Professional wrestling as celebrity

Ford, S. (2017). “He’sa real man’s man”: Pro Wrestling and Negotiations of Contemporary Masculinity. In The Routledge Companion to Media Fandom (pp. 174-183). Routledge.

Ford, S., Sandvoss, C., Real, M., & Bernstein, A. (2007). Mick Foley: Pro wrestling and the contradictions of contemporary American heros. Futures of Entertainment.

Litherland, B. (2018). Wrestling in Britain: Sporting entertainments, celebrity and audiences. Routledge.

Phillips, T. (2015). Wrestling with grief: fan negotiation of professional/private personas in responses to the Chris Benoit double murder–suicide. Celebrity Studies6(1), 69-84.

Ward, D. (2019). ‘Know your role’: Dwayne Johnson & the performance of contemporary stardom. Celebrity Studies10(4), 479-488.

Author Response

I want to thank Reviewer One for their kind remarks about the MS. They were most certainly right about the need to ground professional wrestling in with the broader American context and, more importantly, to do justice to the rich scholarship which exists within this area.

The MS has been added to extensively with engagement with previous texts. Succinctly these additions can be divided into

1) Hilton et al.'s work on the historical development and importance of wrestling in American history

2) Work on kayfabe and anti-fans in particular to discuss the co-creation of kayfabe - or in this case why fans connected with Hogan

3) Broader harmful stereotypes in the WWF/E ranging from IRS to the New Day and work on women.

I do hope these changes satisfy their suggestions. I believe the MS much improved thanks to the previous round.

Reviewer 2 Report

Brief Summary

The author provided an insightful discussion of the representation of Hulk Hogan in professional wrestling in the 1980s in the context of broader social, cultural, and political issues (e.g., Ronald Reagan’s strenuous body politics). For this, the author appropriately used the work of Pierre Bourdieu as a theoretical framework. The application of such concepts as physical capital, symbolic capital, and habitus added to the depth of the scholarly discussion. The author did a very nice job of painting a picture for the readers with rich descriptions of the visual representations of Hogan and other wrestlers. Overall, the discussion fits nicely with prior research and adds to the scholarly work on American physical culture, media representation, and masculinity in the 1980s. 

General Comments

I enjoyed reading this article. The author had a very nice fluid and vivid writing style, which was important for this topic about visual representation. The theoretical and conceptual connections were clear and performed well.

My major critique relates to a mismatch between the title and abstract on one side, and the content of the manuscript, on the other. The title “Whiteness, Masculinity and Americana in Professional Wrestling” and the abstract focus on whiteness. In the actual manuscript, the author only touches on whiteness and race. In order to address this, I see two solutions. First, the author could add discussion related to whiteness. In the specific comments, I suggested several passages where this could be done. Second, would be the option to make the title and abstract match the content of the manuscript. Currently, the author discussed Christianity and its representations much more than race and whiteness. Without changes, I think a title like “Representations of Hulk Hogan in the 1980s: Christianity, masculinity, xenophobia” would be a more fitting for the manuscript.

The overall recommendation regarding acceptance of the article, will depend on what option the author chooses. Expanding the discussion of whiteness, might result in a decision to reconsider after major revisions. An adjustment to the title and abstract, might result in a decision of accept after minor revisions.

Specific Comments

References: According to the journal guidelines, “references may be in any style, provided that you use the consistent formatting throughout.” Currently it appears that the author is not using one citation style consistently. Plus, there are mistakes, for example not inserting a space before the number in parenthetical citations.

Abstract: include significant findings in the abstract. Note that in the current version of the manuscript the author discusses whiteness and heterosexuality very little. Make sure to make the title of the article and abstract match the content of the manuscript. Also see my comment about the conclusion section.

Line 5: use a lowercase t for the word title.

Line 14: include a comma after gender studies.

Line 18: include whiteness as a keyword.

Line 50: Avoid anthropomorphism ("the attribution of human characteristics or behavior to a god, animal, or object”). For example, instead of writing "the textbook discussed," refer to the author(s) instead. Correct this throughout the document. I will not point this out again after this, so you will have to check on your own. 

Line 78: correct spacing.

Line 81: insert centuries in the sentence.

Line 88: consider including a brief statement delineating the development of professional wrestling and sportified wrestling (e.g., in the Olympics).

Line 157: include a citation after “… muscular bodies.”

Line 158: include a citation after “… body guy.”

Line 162: be more precise about the timeframe here. Replace “later” with a specific year or time period. Include a citation here as well.

Line 178: consider a different word for “told.”

Line 184: notice that you have said little about whiteness and raised in general yet. Given the title of the paper, this is surprising. At this point it might also be worthwhile to briefly discuss the role of bodybuilding and physical capital for women in the 1980s.

Line 192: correct spelling of “Ronald Reagan.”

Line 199: consider including information about Reagan’s on college football career, as well as his role of playing George Gipp in the 1940 movie “Knute Rockne.” The letter is significant for the audience of this journal. It also connects nicely to the visual representation of Hulk Hogan.

Line 224: consider exploring the visual juxtaposition of Hulk Hogan’s whiteness and Laurence Tureaud Blackness. This, too, would help the discussion live up the at article’s title.

Line 289: provide a citation to the concept of muscular Christianity. At this point, it might also be worth briefly exploring Christianity and Reaganism in the 1980s.

Line 309: I wonder if “era” would be a less loaded term here than “regime.”

Line 313: coming from behind victories has always been appealing in sports and representations of sport. In the context of the 1980s, however, I wonder if the pattern that you describe here was particularly appealing for Americans. CC my arguments that happen made about the “Miracle on Ice” (the actual performance of the United States men’s ice hockey team in the 1980s Olympics, as well as later movie representations), coming in the footsteps of President Carter’s “crisis of confidence” speech in July 1979. In fact,  I wonder if it could be argued that this underdog/coming-from-behind representation and appeal was part of the American (sport) habitus of the 1980s. Notice possible connections of this discussion with the section on “Hogan Embodiment and Xenophobia.”

Line 338: provide an example and citation for the Hollywood heroics of defeating Soviet counterparts. Again, notice the possible “Miracle on Ice” connection and related literature analyzing the movie adaptation.

Line 341: Consider providing a brief historical context for the US-Iranian relationship in the late 1970s and 1980s with citations. One or two sentences should suffice.

Line 334: As mentioned previously, I think this is an opportunity to further discuss the representation of Hogan’s whiteness in contrast to Vaziri’s appearance.

Line 395: correct spacing in the block quotation.

Line 407: Correct the bolded text.

Line 413: If you want to keep whiteness as a central topic of the manuscript, I think it needs to be discussed in this section again in relation to Kamala v. Hogan. Currently, you discuss Christianity and its representations much more than race and whiteness. Without changes, I think “Representations of Hulk Hogan in the 1980s: Christianity, masculinity, xenophobia” would be a more fitting title for the manuscript.

Conclusion: As discussed thought my comments, note the absence of whiteness and heterosexuality in the conclusion section. Consider returning to the concept of habitus in the conclusion, as it was central to the discussion.

Author Response

I want to thank Reviewer Two for their kind words and encouragement with the Ms. It is, in my view, much improved thanks to their insight. While I haven't been able to include all of the suggestions outside of typos - due in part to Reviewer 1's demands for more engagement with wrestling scholarship, I have included three major changes

1) The focus on whiteness is removed, with more work on Christianity in its place

2) A better contextualisation of Reagan, his administration and US-Iranian relations

3) A discussion of the comeback/underdog habitus in the US at this time - especially in popular culture. This was a genius (!) suggestion and one I cannot believe evaded me initially. This has been added in to expand on Hogan's popularity.

I do hope these revisions will suffice and please accept my sincere thanks.

Round 2

Reviewer 1 Report

Enough of the changes I suggested have been embraced, and I think these add quality to the overall article. Congratulations, I look forward to seeing it in print.

Author Response

Thank you! I really do appreciate all your help